# Detecting flow features in scarce trajectory data using networks derived from symbolic itineraries: an application to surface drifters in the North Atlantic

David Wichmann[1,2], Christian Kehl[1], Henk A. Dijkstra[1,2], and Erik van Sebille[1,2]

[1]Institute for Marine and Atmospheric Research Utrecht, Utrecht University
[2]Centre for Complex Systems Studies, Utrecht University

**Correspondence:** David Wichmann (d.wichmann@uu.nl)

**Abstract.** The basinwide surface transport of tracers such as heat, nutrients and plastic in the North Atlantic Ocean is organized into large scale flow structures such as the Western Boundary Current and the Subtropical and Subpolar Gyres. Being able to identify these features from drifter data is important for studying tracer dispersal, but also to detect changes in the large scale surface flow due to climate change. We propose a new and conceptually simple method to detect groups of trajectories with
similar dynamical behaviour from drifter data using network theory and normalized cut spectral clustering. Our network is constructed from conditional bin-drifter probability distributions and naturally handles drifter trajectories with data gaps and different lifetimes. The eigenvalue problem of the respective Laplacian can be replaced by a singular value decomposition of a related sparse data matrix. The construction of this matrix scales with $O(NM+N\tau)$, where $N$ is the number of particles, $M$ the number of bins and $\tau$ the number of time steps. The concept behind our network construction is rooted in a particle's symbolic
itinerary derived from its trajectory and a state space partition, which we incorporate in its most basic form by replacing a particle's itinerary by a probability distribution over symbols. We represent these distributions as the links of a bipartite graph, connecting particles and symbols. We apply our method to the periodically driven double-gyre flow and successfully identify well-known features. Exploiting the duality between particles and symbols defined by the bipartite graph, we demonstrate how a direct low-dimensional coarse definition of the clustering problem can still lead to relatively accurate results for the most
dominant structures, and resolve features down to scales much below the coarse graining scale. Our method also performs well in detecting structures with incomplete trajectory data, which we demonstrate for the double-gyre flow by randomly removing data points. We finally apply our method to a set of ocean drifter trajectories and present the first network-based clustering of the North Atlantic surface transport based on surface drifters, successfully detecting well-known regions such as the Subpolar and Subtropical Gyres, the Western Boundary Current region and the Carribean Sea.

*Copyright statement.* TEXT

# 1 Introduction

The transport of tracers such as heat, nutrients or plastic in the ocean is an important field of research in oceanography (van Sebille et al., 2018). Despite the inherent time dependence of oceanic transport due to turbulence and temporal variations in the forcing, on the large scale, transport is organized into quasi-stationary regions that are characterized by distinct flow properties. Examples include the five major ocean basins, the Subtropical and Subpolar Gyres, the Western Boundary Currents, etc. Clearly, understanding these features is important for studying the dispersal of tracers. At the same time, changes in external conditions such as through climate change might lead to variations in these large scale flow features (Wu et al., 2012; Beal and Elipot, 2016), and it is therefore important to develop methods that identify and characterize them based on oceanographic data sets.

Many methods exist to detect fluid structures such as regions with little fluid exchange, transport boundaries and coherent structures based on Lagrangian trajectory data (Hadjighasem et al., 2017). While most of these methods are traditionally applied to complete sets of uniformly distributed particle trajectories, recent methods have been successful in identifying coherent structures from incomplete trajectory data (Froyland and Padberg-Gehle, 2015; Padberg-Gehle and Schneide, 2017; Banisch and Koltai, 2017), making them suitable for applications to ocean drifter trajectories with data gaps and different drifter lifetimes. These methods essentially consist of two steps: first, the definition of a measure of similarity $s(n,n')$ or a distance measure $d(n,n')$ that defines how similar or different two trajectories $n$ and $n'$ are; and second, the choice of a clustering algorithm to group trajectories with similar behaviour together. The computational cost and the physical interpretability of trajectory clusters depend on these choices.

Froyland and Padberg-Gehle (2015) embed trajectories in a high-dimensional euclidean space, i.e. they define the distance between trajectories as an abstract euclidean distance (or cosine distance for trajectories on the earth surface) using the entire trajectories, and directly cluster the embedded trajectories with a Fuzzy-c-Means algorithm. Padberg-Gehle and Schneide (2017) define a binary network that indicates if two particles come closer than a certain distance $\epsilon$, together with spectral clustering, see also Banisch et al. (2019). Other methods related to clustering make use of diffusion maps (Banisch and Koltai, 2017) or the dynamical distance of trajectories (Hadjighasem et al., 2016). The latter three methods all use a spectral relaxation of the normalized cut problem (NCut) for the trajectory classification, which was introduced by Shi and Malik (2000).

Here, we propose a new and conceptually simple network-based method to identify groups of trajectories that have similar dynamical behaviour. The network is constructed based on ideas from symbolic dynamics, which describes the coarse grained trajectory of a particle given some partition (binning) of the state space. The itinerary of a particle, i.e. the sequence of bins it visited, if known for long times, resolves information much below the bin resolution. Different from previous network-based methods, we make full use of the duality between individual particles and their coarse grained itineraries, which can lead to significant computational advantages. Here, we simplify the itineraries to a minimum: neglecting the time dimension, we represent the trajectory data as a bipartite network connecting particles and bins, with links defined by conditional distributions over symbols. With an appropriate choice of similarity measure, our method allows us to formulate spectral relaxations of the NCut (Shi and Malik, 2000) in terms of the singular value decomposition (SVD) of a related data matrix. Setting up this matrix

scales with $O(NM + N\tau)$, $N$ being the number of trajectories, $M$ the number of bins and $\tau$ the number of time steps. Our method is naturally extendable to incomplete trajectory data and thus readily applicable to ocean drifters. Conceptually, our method is close to detecting minimally mixing fluid regions, so called almost-invariant sets (Froyland, 2005), although there are also important differences to this method, cf. section 3.4. In our case, almost-invariant sets are represented by the initial conditions of groups of particles, with trajectories of different groups having only little overlap.

We show with a model flow, the periodically driven double-gyre flow, that the method accurately finds almost invariant regions and transport barriers. We also show that the method correctly classifies most of the trajectories in an incomplete data set. Our method can also be used to formulate the clustering problem in a low-dimensional setting by choosing a coarse partition, which can still resolve the leading order flow structures to a high accuracy. We show this for the double-gyre flow with an effectively 9-dimensional formulation of the clustering problem, which still resolves the leading features of the flow up to details much

below the coarse graining scale. Our method is designed for detecting quasi-stationary features from ocean drifter trajectories. We therefore emphasize that, owing to the strong simplifications of the particle itineraries, the method is not suitable for the detection of coherent vortices that are transported in a background flow, such as the 'Bickley Jet' (discussed e.g. in Hadjighasem et al. (2017)).

Several trajectory based methods have been applied to the ocean drifter data set on the global scale (Froyland and Padberg-

Gehle, 2015; Banisch and Koltai, 2017). In addition, transfer operator methods based on virtual particle trajectories have been used to detect almost-invariant sets at the ocean surface (Froyland et al., 2014). While these methods successfully identified the five major ocean basins, each of these basins has an approximate attractor in its center (Froyland et al., 2014; Wichmann et al., 2019), such that the long-term dynamics of global drifter trajectories is exceptionally low-dimensional. Using the ocean drifter data set of the Global Drifter Program (Lumpkin and Centurioni, 2019), we apply our method to identify prominent flow

features in the North Atlantic Ocean only, and present here the first drifter based clustering result of the North Atlantic surface flow using network theory. Using data from around 8,300 drifter trajectories, we successfully detect well-known features such as the boundary between the Subtropical and Subpolar Gyres as discussed in Brambilla and Talley (2006), the Western Boundary Current, the Carribbean Sea and the separation between the Subpolar Gyre and the Nordic Seas (Bower et al., 2019).

## 2   Drifter data set

We use daily drifter data derived from the six-hourly interpolated data from the NOAA-AOML global drifter program (Lumpkin and Centurioni, 2019), and constrain the data set to those drifters that were released (but not necessarily stay) in the North Atlantic. The restriction to the North Atlantic leaves us with 5,270 drifter trajectories, starting from 1989. A major challenge in analysing the data set is its inhomogeneity in space and time. Figure 1a shows the distribution of drifter locations along all trajectories with a square binning of $2°$ in both longitude and latitude. It is visible that most of the data is located in the centre

of the basin, the subtropical gyre. The accumulation of drifters in the gyre centres is a well-known feature of the basin-scale surface ocean, and is attributed to Ekman convergence in the centre of the gyre, also explaining the accumulation of marine debris in these areas (Kubota, 1994; van Sebille et al., 2020). Figure 1b shows the distribution of release locations, again with

a square binning of $2°$. Drifters are mostly released in regions with strongly repelling properties in order to probe the flow that is rarely sampled in the long term. Most notably, many drifters are released in the Western Boundary Current region (along the US east coast), including the Gulf Stream with vigorous mixing and strong currents. Drifter lifetimes (fig. 1c) also vary widely, which poses a challenge if trajectories of different lengths need to be compared. Our algorithm is set up in a way that it naturally handles trajectories of different release times, lengths and lifetimes such that all drifter trajectories are available for our analysis, cf. section 3.

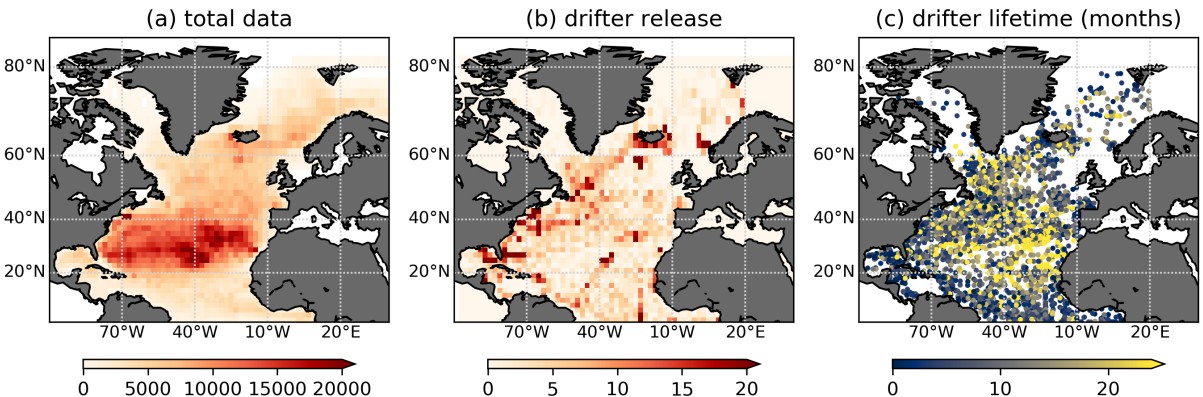

**Figure 1.** a: Total counts of all drifter locations (6-hourly for each trajectory), computed with a binning of $2°$ in both longitude and latitude. b: Distribution of drifter release per $2°$-bin. c: Scatter plot of initial drifter location, the colour indicating the drifter lifetime in months.

## 3  Methods

### 3.1  Preliminaries

Suppose we are given a set of $N$ drifter trajectories at $\tau$ time instances $x_n(t), n = 1, \ldots, N, t = 0, \ldots, \tau - 1$. We divide the fluid domain $\Omega$ into $M$ disjoint sets (bins) $\{B_m\}$ such that $\cup_m B_m = \Omega$. Given such a partition, a particle trajectory can be described by a symbolic sequence of bin labels $m = 1, \ldots, M$, called *itinerary*, which is a representation of the trajectory in terms of symbolic dynamics, see fig. 2 for an example. The bin labels $m = 1, \ldots, M$ are called the *alphabet* of the symbolic dynamics. We define the coarse grained binary position vector $\delta_n(t) \in \{0,1\}^M$ of particle $n$ as $\delta_{n,m}(t) = 1$ if $x_n(t) \in B_m$, and $\delta_{n,m}(t) = 0$ otherwise. If the data set is incomplete in the sense that for some particles $n$, $x_n$ is not defined at time $t$, we simply set $\delta_{n,m}(t) = 0$ for all $m$. As the sets $B_m$ are disjoint, $\delta_n(t)$ has maximally one non-zero entry at any point in time. With this definition, we define the coarse grained data matrix $C(t) \in \mathbb{R}^{N \times M}$ as

$$C_{nm}(t) = \delta_{n,m}(t). \tag{1}$$

As a function of time, the matrices $C(t)$ describe the coarse-grained dynamics of the entire data set. Note that $C(t)$ is very sparse, with maximally $N$ non-zero entries at any time $t$. Next, we define a matrix $G$ as:

$$G = \sum_{t=0}^{\tau-1} C(t). \tag{2}$$

     The matrix $G$ has a simple interpretation: $G_{nm}$ is equal to the number of times that particle $n$ visited bin $m$ at the time instances $0, 1, \ldots, \tau-1$. $G$ defines a bipartite graph, i.e. a graph connecting objects of different type, here particles and bins. In

the following, we define the degree vector $d[A] \in \mathbb{R}^h$ of an arbitrary matrix $A \in \mathbb{R}^{h \times l}$ by $d[A]_i := \sum_{j=1}^l A_{ij}$, and the degree matrix $D[A] \in \mathbb{R}^{h \times h}$ as $D[A] := \text{diag}(d[A])$.

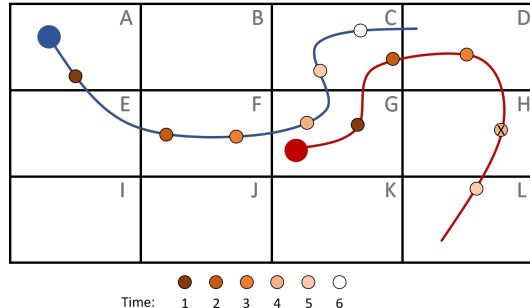

**Figure 2.** Two example trajectories on a binned fluid domain, with alphabet $A, \ldots, L$ and $\tau = 7$. The markers along the trajectories represent the points in time the location is stored. The itinerary for the blue particle is: $AAFFGCC$. The red particle has itinerary $GGCD\_L\_$. Here, '_' denotes a missing data point, the one at time $t = 4$, indicated by an 'X' in the respective marker. The last data point of the red trajectory is also missing, as its lifetime is shorter than $\tau$ time instances.

## 3.2    Definition of the network

     Given a partition $\{B_m\}$, the itineraries of a group of particles can be used to slice the state space into smaller regions. For example, all particles having the same itinerary for a certain number of time steps could be grouped together. With increasing

trajectory length, this partitions the state space into smaller and smaller sets of initial conditions that have similar dynamic behaviour. We refer to chapter 14 of the open source book by Cvitanović et al. (2016) for an introduction to symbolic dynamics and the use of particle itineraries to partition the state space. In applications, requiring exactly equal itineraries is not very practical. This is because in a chaotic flow two particles that start close may separate exponentially. As the number of all possible itineraries is still very large - there are $M^\tau$ of them - there would be no two equal itineraries after a short time.

We therefore define another continuous similarity measure $s(n, n')$ for two particles $n$ and $n'$, with $0 \leq s(n, n') \leq 1$, based on the particles' itineraries. For clarity, we will write the itineraries in terms of letters rather than numbers, and imagine that each symbol is part of an alphabet with $M$ letters. Missing data is represented by '_'. We require the following properties, using an example itinerary $AABC$:

(Req. 1) Invariance to permutation: $s(AABC, CABA) = s(AABC, AABC) = 1$

(Req. 2) Sensitivity to missing data: $s(AABC, \_ABC) < 1$, $s(AABC, AA\_C) < 1$, $s(\_ABC, \_ABC) < 1$, $s(AABC, \_ABC) = s(AABC, DABC)$

(Req. 3) Zero similarity for disjoint itineraries: $s(AABC, DEFF) = 0$.

Requirement 1 essentially discards the time dimension. Requirement 2 takes into account that an itinerary with data gaps contains less information than a full itinerary, and that a missing data point should be treated just as another symbol ('D') that is not part of the example itinerary. Requirement 3 states that completely different itineraries have zero similarity. The easiest way to implement requirements 1-3 is through introducing a conditional symbol distribution $p(m|n) \in \mathbb{R}^M$, defined for each particle $n$ and symbol $m$, and an appropriate choice of similarity measure between these distributions. We define the distribution by normalization of the individual symbol counts with the *total* trajectory length, i.e. including data gaps. For example, a particle $n$ with itinerary 'AB_FCCH' has

$$p(A|n) = p(B|n) = p(H|n) = p(F|n) = 1/7, \; p(C|n) = 2/7. \tag{3}$$

All requirements are fulfilled with the similarity measure

$$s(n, n') = \sum_m p(m|n)p(m|n'). \tag{4}$$

Identifying each letter in the symbolic alphabet with a number $m = 1, \ldots, M$, we can directly relate $s(n, n')$ to the matrix $G$ defined in eq. (2) as $s(n, n') = \frac{1}{\tau^2} \sum_m G_{nm} G_{n'm} = \frac{1}{\tau^2}(GG^T)_{nn'}$. For spectral clustering with the normalized cut (NCut), the constant $\frac{1}{\tau^2}$ is irrelevant. Thus, we define the following network of similarities on particle trajectories:

$$Q := GG^T \in \mathbb{R}^{N \times N}. \tag{5}$$

This is the projection of the bipartite network $G$ onto particle space, see e.g. section 10.4 of the book by Fouss et al. (2016). Note that in the case of two non-mixing (invariant) sets, for example as in the autonomous double-gyre flow discussed by Froyland and Padberg (2009), network $Q$ defined in eq. (5) can be brought into block-diagonal structure. This can be achieved with a partition that is optimal in terms of the invariant sets, i.e. if each set can be completely covered by a part of the alphabet. For non-optimal partitions, we expect that it is still possible to detect an imprint of the two invariant sets with a clustering algorithm, which we quickly present in the following section.

### 3.3 Normalized cut and spectral relaxation

In this section we sketch the method of solving a relaxed version of the NCut according to Shi and Malik (2000). Our methods are equal to the simultaneous K-way NCut method described in Von Luxburg (2007) and the hierarchical clustering of Shi and

Malik (2000). The main difference is that our network defined in eq. (5) allows to compute an SVD instead of solving the eigenvalue problem of the Laplacian.

Assume we are given an undirected network defined on a discrete set $S$ containing $N$ vertices, with edges given by the symmetric adjacency matrix $Q \in \mathbb{R}^{N \times N}$. We assume that $Q$ is connected. If it is not connected, we focus on each connected component separately. According to Shi and Malik (2000), the normalized cut of a partition of the nodes into $K$ sets $S_1, \ldots, S_K$, $S = \cup_{i=1}^{K} S_i$, is defined as

$$\text{NCut}(S_1, \ldots, S_K) := \sum_{i}^{K} \frac{Q(S_i, S_i^C)}{Q(S_i, S)}. \tag{6}$$

Here, $Q(S_i, S_j)$ is the sum of all weights connecting $S_i$ and $S_j$, i.e. $Q(S_i, S)$ is the sum of all weights connected to $S_i$. $S_i^C$ denotes the complement of $S_i$. The term $\frac{Q(S_i, S_i^C)}{Q(S_i, S)}$ appearing in the definition of the NCut in eq. (6) is simply the total weight of all the edges connecting a set $S_i$ to its complement relative to the total weight of the set $S_i$. Clustering a graph according to the NCut refers to finding a partition $\{S_k\}$ such the objective function in eq. (6) is minimized. Note that for an increasing number of sets $\{S_k\}$, the NCut can never decrease. Minimizing the NCut leads to a clustering result that tries to balance the different terms in eq. (6), such that the resulting clusters are of approximately equal size in terms of their total relative weight (Shi and Malik, 2000). While this poses no serious problem for detecting large scale flow features in the ocean, it is certainly a limitation for detecting even smaller structures such as eddies or jets in a large ocean domain, and we explicitly exclude such examples from the scope of our method.

As shown in Shi and Malik (2000), an approximate solution to the problem can be constructed using the eigenvectors of the *symmetric normalized Laplacian* of $Q$, defined by

$$L_s[Q] = \mathbb{I} - D[Q]^{-1/2} Q D[Q]^{-1/2}. \tag{7}$$

Such a solution is only approximate, as constraints of the optimization problem are neglected, hence the term 'spectral relaxation', see also Fan and Pardalos (2012) for a discussion of different relaxations of the NCut. $L_s[Q]$ is positive-semidefinite and its eigenvalues $\{\lambda_{s,i}\}$ satisfy $\lambda_{s,0} = 0 < \lambda_{s,1} \leq \ldots \lambda_{s,N-1}$. To identify clusters with the NCut under spectral relaxation, we need the eigenvalues and the right eigenvectors, $\{\lambda_{r,i}, v_{r,i}\}$ of the random walk Laplacian (Shi and Malik, 2000)

$$L_r[Q] = D[Q]^{-1} Q, \tag{8}$$

which are related to the spectrum of $L_s[Q]$ in the following way:

$$\begin{aligned} \lambda_{r,i} &= 1 - \lambda_{s,i}, \\ v_{r,i} &= D[Q]^{-1/2} v_{s,i}. \end{aligned} \tag{9}$$

To find $K$ clusters in $Q$ with the NCut under spectral relaxation, one can either apply a hierarchical clustering procedure as done by Shi and Malik (2000), or a simultaneous K-way cut, see Von Luxburg (2007) for more information. From a compu-
tational perspective, in our case the K-way cut is preferable as we can compute the spectrum of the Laplacian directly from the SVD of a related matrix. This is because for the normalized cut with spectral relaxation, we need the eigenvectors corresponding to the $K$ largest eigenvalues of $D[Q]^{-1/2}QD[Q]^{-1/2}$. As for any two matrices, $d[AB] = Ad[B]$, it follows that these eigenvectors are equal to the left singular vectors corresponding to the $K$ largest singular values of the matrix

$$R := \operatorname{diag}(Gd[G^T])^{-1/2}G. \tag{10}$$

Equation (10) also motivates to consider the right singular vectors of $R$. These are indeed under certain conditions related to the almost-invariant sets based on the transfer operator according to Froyland (2005), see appendix A.

The algorithm for the simultaneous K-way cut for $Q$ in our case is:

**Algorithm 1: simultaneous K-way clustering**

S1  Compute the first $K$ left singular vectors of $R = \operatorname{diag}(Gd[G^T])^{-1/2}G$, $v_{s,0}, \ldots, v_{s,K-1}$.

S2  Compute $v_{r,i} = D[Q]^{-1/2}v_{s,i}$ for $i = 0, \ldots, K-1$.

S3  Embed the $N$ nodes of the network in $\mathbb{R}^K$ by setting $y_n = (v_{r,0,n}, v_{r,1,n}, \ldots, v_{r,K-1,n})$, $n = 1, \ldots, N$.

S4  Perform a standard euclidean-space clustering algorithm (here k-Means) on the $N$ points $y_n \in \mathbb{R}^K$.

We choose this algorithm for the double-gyre flow (cf. section 4.1) for comparison with previous methods such as Padberg-
Gehle and Schneide (2017), Banisch and Koltai (2017) and Hadjighasem et al. (2016). For the ocean drifter data set (cf. section 4.2), we choose a hierarchical method instead of the simultaneous K-way cut, following Shi and Malik (2000). The reason is that the hierarchy preserves the most important boundaries of a clustering solution. This is physically desirable, as several of these main boundaries are known to oceanographers. In addition, it simplifies the presentation of our results and different choices of $K$ when combined with a dendrogram. Note also that if a network is initially disconnected, as is the case for the
North Atlantic drifters, it is necessary to compute $Q$ in order to determine the individual connected components.

**Algorithm 2: hierarchical clustering**

H1  Compute the network $Q$ defined in eq. (5).

H2  Find the largest connected component of the network and restrict $Q$ to it. Define $\bar{Q}$ as this restriction.

H3  Compute the eigenvector $v_{s,1}$ of $L_s[\bar{Q}]$.

H4  Compute $v_{r,1} = D[\bar{Q}]^{-1/2}v_{s,1}$

H5 Find a cutoff $c$ such that the sets of nodes defined by $S_1 = \{n \in \{1, \ldots, N\} \mid v_{r,1,n} < c\}$ and $S_2 = \{n \in \{1, \ldots, N\} \mid v_{r,1,n} \geq c\}$ minimize the NCut for the two sets $S_1$ and $S_2$.

H6 Split the original network into two networks, with respective adjacency matrices $Q_{S_1}, Q_{S_2}$ defined by the projection of $\bar{Q}$ onto these sets.

H7 For each sub-network, repeat steps H3-H5.

H8 Choose to split the sub-network that minimizes the generalized normalized cut in eq. (6).

H9 Repeat the steps H3-H8 up to a certain number of sets $K$.

At each iteration, only one of the clusters is split into two, and the *global* NCut is minimized at each step. This is different from other hierarchical procedures such as in Ma and Bollt (2013), where a local coherence ratio is maximized and the number of clusters can double at each iteration. Algorithm 2 was, apart from H2, suggested by Shi and Malik (2000) and is preferable if we do not have a specific bound for the coherence of an individual cluster in mind, ensuring to have, at each step, the most important clusters in terms of eq. (6). Note that we do not check for each sub-network if it is connected, as the sub-networks derived from the North Atlantic drifters were connected (only one eigenvalue equal to zero). However, this might be necessary for other data sets. We also apply algorithm 2 to the double-gyre model flow to see how the clustering result is affected by the choice of algorithm.

Note that there is no general rule to determine the number of clusters $K$ in algorithm 1, or where to stop the hierarchical clustering procedure in algorithm 2. A popular heuristic to determine $K$ is to look for a prominent gap in a spectrum of $L_s$ and choose $K$ as the number of smallest eigenvalues before that gap (Hadjighasem et al., 2016). This is however problematic for systems with no prominent spectral gap, which is the case for the systems considered here (cf. section 4). For algorithm 1, we therefore compute the clustering results for different values of $K$ to see how the results depend on this choice. For algorithm 2, one can set a maximum value on the cost function in eq. (6) as suggested by Shi and Malik (2000). Yet, the cost function is rather abstract, and there is no general rule what this value should be. Here, we compute the clusters up to a certain (arbitrary) order and compare the results to known structures in oceanography.

## 3.4 Comparison to existing methods

Our method aims to detect groups of particles, with trajectories of different groups having only little overlap. In this sense, our method detects groups of particles with little mixing between each other, which is close to detecting almost-invariant sets according to Froyland (2005). Yet our method is different from detecting almost-invariant sets with the transfer operator in several aspects. First, it is based on similarities between individual particles rather than spatial sets (bins), which allows us to cluster on the particle level rather than the bin level. As we will show in section 4.1, this can be used to resolve flow features down to scales much below the bin size. Secondly, our method employs the full trajectory information in terms of a particle's symbolic itinerary, rather than just the start and end points or symbols. In practical applications, this can be an advantage compared to the transfer operator framework, as there is no need in assuming Markovian behaviour of the flow given a state

space partition, as done by e.g. Froyland et al. (2014).

There are also major differences between our method and other existing methods that cluster on the particle level (Froyland and Padberg-Gehle, 2015; Padberg-Gehle and Schneide, 2017; Banisch and Koltai, 2017; Hadjighasem et al., 2016). First, these methods only compare particles at equal times, while we disregard the time information. This can be a significant advantage in situations where the major features of a flow are approximately stationary, i.e. can be seen as part of a (noisy) autonomous dynamical system. In this case, using the time information of drifter trajectories should not be necessary. Especially for the

ocean drifter dataset, containing drifters of different starting times and lengths, it would be very difficult if not impossible to find sub-basin large scale structures when restricting to drifters that necessarily overlap temporally, although this is possible on the global scale to identify the basins themselves (Froyland and Padberg-Gehle, 2015; Banisch and Koltai, 2017). Note that simply placing all drifters at the same initial time and proceeding with one of the existing methods would lead to further problems, as there is ambiguity in which point of a trajectory should be taken for the initial time, probably requiring more data

processing such as demanding an initially uniform particle distribution. Our method of simplifying the trajectories does not have these problems by construction, and can be readily applied to scarce drifter datasets.

From a computational perspective, setting up one of the sparse matrices $C(t)$ in eq. (1) is $O(N)$ such that computing the matrix $G$ in eq. (2) is $O(N\tau)$. Computing $d[G^T]$ is $O(NM)$ as is the product $Gd[G^T]$. In total, computing $R$ of eq. (10) is therefore of computational complexity $O(NM+N\tau)$. If we work with $R$ directly, i.e. we use the simultaneous K-way clustering method

described in algorithm 1 in section 3.3, computing this network is of lower computational complexity as the computation of the networks used by other studies (Padberg-Gehle and Schneide, 2017; Banisch and Koltai, 2017; Hadjighasem et al., 2016). These methods typically rely on comparing particle positions between all particles at all time instances, i.e. they scale with $O(N^2\tau)$ in the worst case, although a nearest-neighbour search as applicable to the studies of Padberg-Gehle and Schneide (2017) and Banisch and Koltai (2017) can reduce the $N^2$ term to something like $N \log N$. Further, the matrix $R$ in eq. (10) is

sparser than $L_s[Q]$ or can have column dimension (= number of bins $M$) significantly lower than row dimension (= number of particles $N$), cf. section 4.1. In these cases, computing the SVD of $R$ instead of the eigenvectors of $L_s$ can lead to computational speed up. Finally, it is interesting to note that the computation of the network is faster for coarser partitions, i.e. when particles are connected in the network even when their trajectories are far apart, as the number of bins $M$ decreases. This is opposite to the methods of Padberg-Gehle and Schneide (2017) and Banisch and Koltai (2017), where computing the network becomes

more costly for larger spatial scale parameters (called $\epsilon$ in both studies).

The major drawback of our method is the dependence on a reference frame with respect to which the phase space partition and thus the symbolic itineraries are defined. This can be understood when imagining a time-independent flow from a rotating reference frame. The rotation of the reference frame contributes to a particle's itinerary, and, by averaging over different points in time, non-zero similarities between trajectories can result from the sole rotation of the reference frame. Due to this reason,

our method can not be applied to strongly time-dependent systems such as the Bickley jet model flow where coherent vortices are transported in a periodic background flow. It is, however, still possible to detect transport boundaries in time-dependent flows such as the periodically driven double-gyre flow, as we show in section 4.1, where particle trajectories belonging to different invariant sets can still be distinguished with a fixed partition.

## 4 Results

### 4.1 Periodically driven double-gyre flow

To test our method, we choose a model flow that has been used for the detection of coherent structures before (Froyland and Padberg, 2009; Froyland and Padberg-Gehle, 2015; Banisch and Koltai, 2017). The periodically driven double-gyre flow is defined on a domain $\Omega = [0,2] \times [0,1]$, with equations of motion:

$$
\begin{aligned}
\dot{x} &= -\pi A \sin(\pi f(x,t)) \cos(\pi y), \\
\dot{y} &= \pi A \cos(\pi f(x,t)) \sin(\pi y) \frac{df}{dx}(x,t),
\end{aligned}
\tag{11}
$$

where $f(x,t) = \epsilon \sin(\omega t) x^2 + (1 - 2\epsilon \sin(\omega t)) x$, and $A = 0.25$, $\epsilon = 0.25$ and $\omega = 2\pi$. Similar to Banisch and Koltai (2017), we initially place 20,000 particles on the vertices of a uniform grid on the domain $(0,2) \times (0,1)$ and compute trajectory outputs for twenty gyre periods with time steps of $0.1$, i.e. we have $\tau = 201$. The eigenvectors are computed with the SVD of the matrix $R$ in eq. (10). These eigenvectors are then used for the K-way clustering algorithm with k-Means, cf. algorithm 1 in section 3.3. Figure 3 shows the result for the clustering of $Q$, plotted at $t = 0$ and a binning of $\Delta x = \Delta y = 0.04$, i.e. the column dimension of $R$ is $M = 1,250$. In the figure, we include higher order (larger $K$) splits as well to show the full range of results obtained by our algorithm. For $K = 2$ (fig. 3b, and all even values of $K$), we see a clear separation between the left and right gyres, which is a common feature found by other studies (Froyland and Padberg, 2009; Froyland and Padberg-Gehle, 2015; Banisch and Koltai, 2017). The figure also resolves to a certain accuracy the expected transport barrier for the blue and brown particles (the filaments extending into the respective other set). For $K = 3$, we separate the gyre centres from their surrounding. Subsequent uneven values of $K$ further split up the gyre centres in slices. Corresponding results for the autonomous double gyre ($A = 1$, $\epsilon = 0$) illustrating the idea of optimal partitions (cf. section 3.2) are shown in figs. B1 and B2 in the appendix.

We emphasize that the clustering of $Q$ in fig. 3 does not show any binning structure. Intuitively, a long itinerary can resolve small-scale structures, i.e. distinguish particles that are initially much closer than the typical bin size, see chapter 14 of Cvitanović et al. (2016). The computational cost can be reduced even more if we make the bins larger. Figure 4 shows the result of the clustering for $\Delta x = 2/3$, $\Delta y = 1/3$, i.e. $M = 9$, up to $K = 4$. Note that the choice of $\Delta x$ prevents a preferable binning along the $x = 1$ line. For this choice, the matrix $Q$ has column dimension equal to nine, i.e. the clustering problem is effectively 9-dimensional. The corresponding clustering result still resolves the most dominant structure up to very high resolution: the split between the left and right side is preserved, and even the structure of the transport barrier is very similar to the one in fig. 3. The results for $K = 3$ and $K = 4$ are however different from fig. 3, as the gyre centres appear smaller. This indicates that only the most prominent structures, here the separation between the left and right sides, are preserved under coarsening the partition. Nevertheless, figs. 4c-d do still give an impression about the flow structures at higher orders, though not completely equal to the high resolution case. Note that the singular values (fig. 4a) are strongly suppressed compared to the $M = 1,250$ case in fig. 3.

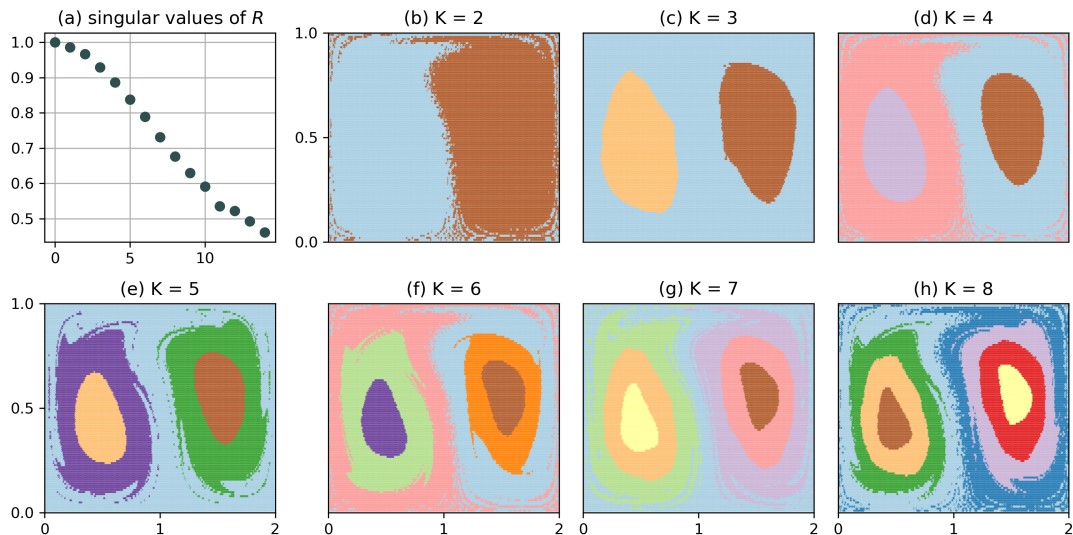

**Figure 3.** Clustering of $Q$ with $\Delta x = \Delta y = 0.04$ of the full data set, i.e. $N = 20,000$ and $\tau = 201$.

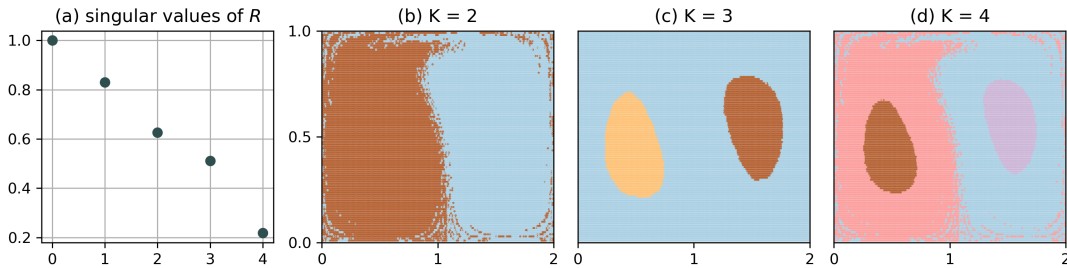

**Figure 4.** Clustering of $Q$ with $\Delta x = 2/3$, $\Delta y = 1/3$ of the full data set, i.e. $N = 20,000$ and $\tau = 201$. The most dominant structure is still visible to high accuracy, although the problem is effectively 9-dimensional.

To test the robustness of our method to missing data, we randomly choose 500 out of the 20,000 particles and for the remaining data set randomly delete 80 % of the data points. This is similar to the approach of Banisch and Koltai (2017) and Froyland and Padberg-Gehle (2015). When the data becomes sparser, different nodes become disconnected, which leads to small, noisy clusters that can be identified by multiple singular values equal to 1. We can remedy that by increasing the bin size such that the network becomes connected again. Therefore, we choose $\Delta x = \Delta y = 0.4$. In doing so, we effectively extend the domain in the $y$-direction to $y = 1.2$ and disregard the fact that the top row of bins is not completely covered with initial conditions. Figure 5 shows the result for the clustering of $Q$ for the incomplete data set, plotted on top of the corresponding clustering result of the full trajectories (i.e. $\Delta x = \Delta y = 0.04$). The result of the incomplete data set roughly agrees with the expected result of the full clustering. For the results shown in fig. 5b, 10 out of the 500 labels were assigned incorrectly compared to the full data case computed with $\Delta x = \Delta y = 0.04$ (fig. 3), see fig. B3 in the appendix for the number of wrongly

assigned labels for different bin sizes. Note again the strong suppression of the singular values for the incomplete data case (fig. 5a).

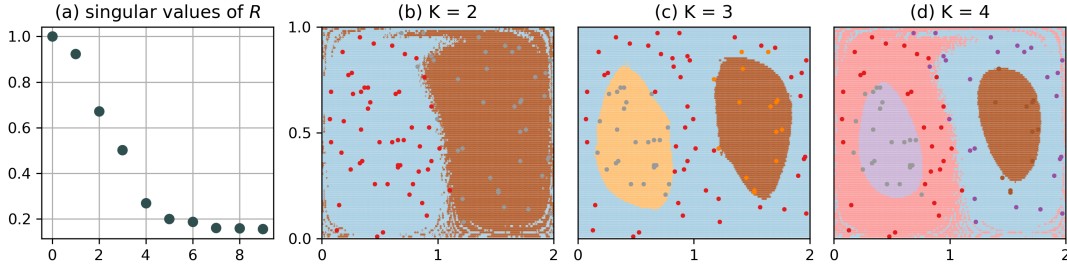

**Figure 5.** Clustering of $Q$ for the incomplete data set, where 500 particles are retained and subsequent deletion of 80 % of the data points. a: singular values of $R$ for the incomplete data set, with $\Delta x = \Delta y = 0.4$. b-d: points representing the initial position of those particles that have data at initial time. Background: result for the full trajectory with $\Delta x = \Delta y = 0.04$ (cf. fig. 3).

    The results for the double-gyre flow illustrate the robustness of our method in identifying the most dominant structures with incomplete trajectory data. Having control over the bin size enables to tune the network such that it stays connected and the major structures can be resolved. At the same time, small-scale features of the flow seem to be resolved, at least to some extent,

independent of the bin size using the K-way simultaneous NCut.

We also tested the algorithm for shorter trajectories, cf. fig. B4 in the appendix, showing an expected change of the boundary filaments between the left and right sides of the fluid, which mix less in the shorter period of time. To better understand the differences between algorithms 1 and 2 introduced in section 3.3, we also applied the hierarchical NCut method to the non-autonomous double gyre flow. A problem arises here for the first split into two clusters, as there is no unique minimum in the

objective function in eq. (6), i.e determining the cutoff $c$ in algorithm 2, H5 (cf. section 3.3), is ambiguous for $\Delta x = \Delta y = 0.04$. The lack of a unique minimum for the NCut has been observed before for the same model flow by Froyland and Padberg (2009) in the transfer operator framework (see their fig. 15) corresponding to the lack of a unique maximum of the coherence ratio there (see proposition 1 in appendix A). Yet, for $\Delta x = \Delta y = 0.04$, in our case, the split between the left and right sides along the transport boundary (the $K = 2$ split in fig. 3) does not even have a local minimum for the NCut, cf. fig. B5, as opposed to

the local maximum of Froyland and Padberg (2009). This however changes to a local minimum for $\Delta x = \Delta y = 0.1$ (fig. B6) and finally to a global minimum for $\Delta x = \Delta y = 0.2$ (fig. B7). A possible explanation of this dependence on bin size is that the addition of noise (i.e. larger bins) decreases the coherence of the gyre centres compared to the transport boundary, making the latter easier to be detected. Due to the sensitivity of the hierarchical clustering result to the bin size, we test different bin sizes for the clustering of the North Atlantic drifters in section 4.2 (cf, fig. C2).

**4.2   Surface drifters in the North Atlantic**

We compute the matrix $G$ for the drifter data set with a square binning of $1°$ and a time step of one day and maximum time of 365 days, i.e. $\tau = 365$. The binning is the same as the one chosen by van Sebille et al. (2012) for the computation of

drifter-derived transition matrices. We discard the time dimension for the trajectories and place every trajectory at $t = 0$ at the location of drifter release. All trajectories exceeding one year are cut into smaller pieces, each of length smaller or equal to one year, which expands our data set from 5,270 to 8,334 trajectories. The network defined by the drifters is not connected, such that we identify the largest connected component prior to the clustering (with the python networkx package). This restriction removes seven out of the 8,334 trajectories, one of which is a short trajectory of nine time steps at the Strait of Gibraltar, and the other six corresponding to invalid drifter data. Note that the size of the largest connected component is expected to decrease with decreasing bin size, as trajectory overlaps are less likely for smaller bins. We cluster the data with the hierarchical NCut algorithm for $Q$, cf. section 3.3.

Figure 6a shows the particle labels at $t = 0$. The dendrogram in fig. 6c shows which groups of particles are split, the y-axis corresponding to the value of the global NCut before the split. See the caption of fig. 6 for the oceanographic names of the different regions. Nine out of the 20 clusters consisted of one or two particles only. These are coloured in grey in the figure, and have a label '.' in the dendrogram.

The first major split separates the Subpolar Gyre and Nordic Seas from the subtropical and tropical North Atlantic. This splits essentially the Subpolar Gyre from the Subtropical Gyre, which compose together a double-gyre system, having some similarity to the one in section 4.1. The scarce transport of drifters between these two regions has been studied before (Brambilla and Talley, 2006; Rypina et al., 2011; McAdam and van Sebille, 2018), and it is promising that we identify this separation first. Compared to Rypina et al. (2011), the separation is slightly shifted northwards at the western side of the basin, the Western Boundary Current region particles, labelled 'E', extending slightly into the Labrador Sea. This changes if we plot the drifters at their final time, see fig. 6b, showing that some particles of group 'E' from the western Labrador Sea are transported into the northern Western Boundary Current region. Note here that the trajectories in the different clusters do have small but non-zero overlap, such that the spatial extent of the clusters can be different at initial and final time.

Next, our algorithm separates the Subpolar Gyre from the Nordic Seas. A relatively clear cut is seen along the Iceland-Scotland ridge. The strength of the transport over the ridge is in fact an old topic in oceanography (Bower et al., 2019). Compared to fig. 6a, this separation is slightly less prominent in figure 6b, indicating some slow but non-zero flow across the ridge. The next separation that our algorithm detects is between the southern and northern parts of the Subtropical Gyre, which can be explained by the slow clockwise rotation of the gyre. The next split separates out the Caribbean Sea (label 'F'), a region well known to be rich of eddies and able to trap water masses, after which the Western Boundary Current region (label 'E') extends northwards along the east coast of the US into the northern part of the subtropics. We eventually also identify the northern part of the Subpolar Gyre (Irminger Current), and separate the Barents Sea from the Norwegian Sea, as well as the Bay of Biscay as last separation of the hierarchical clustering algorithm.

We also tested our clustering algorithm without constraining the trajectory length of the data set, see fig. C1 up to comparable values of the NCut. We still resolve many of the major features of the flow in the North Atlantic such as the Western Boundary Current region and the Caribbean Sea, the Subpolar Gyre and the Norwegian Sea. Others disappear, e.g. the Bay of Biscay and the Irminger Current. This is likely an effect of longer trajectories receiving more weight in the definition of our network, cf. section 3.2, or due to the fact that some particle groups mix with each other on longer time scales, cf. section 4.1. It could,

however, also be an effect of the fewer trajectories compared to the case when we set a cutoff on trajectory length. For example, there is almost no drifter starting in the Barents sea in the full data set, see fig. C1.

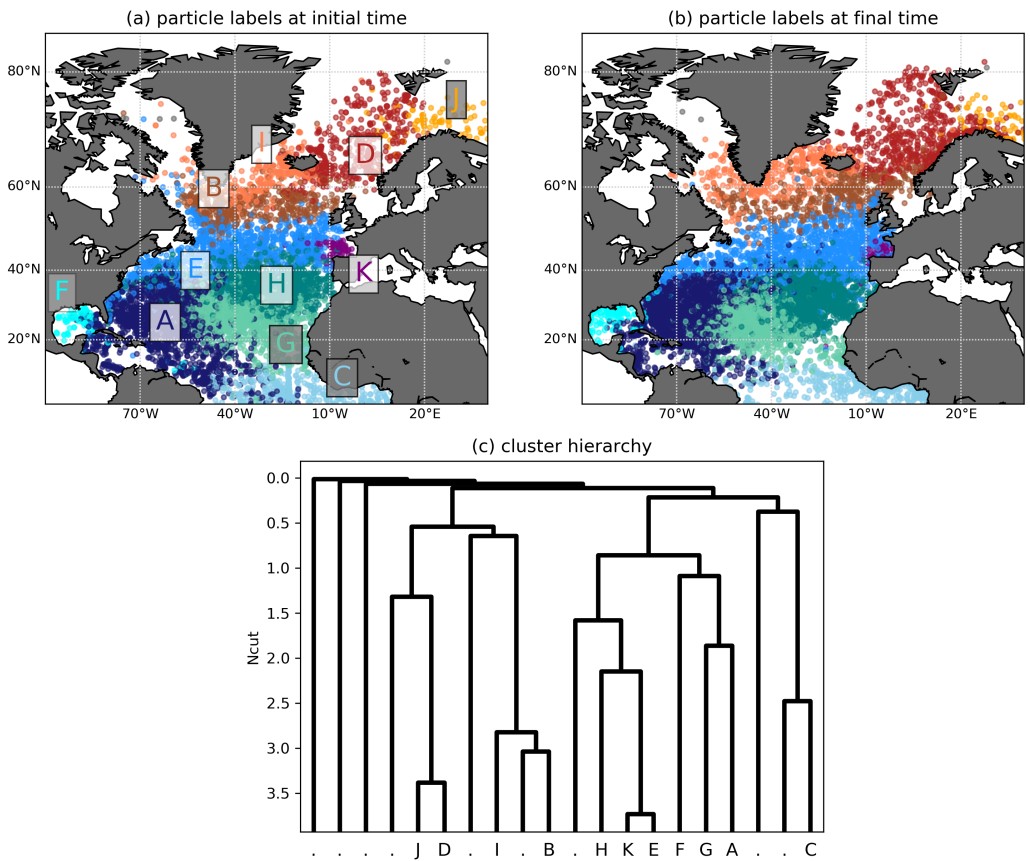

**Figure 6.** a: clustering of $Q$ for the drifter data set with $\Delta x = \Delta y = 1°$. Clusters corresponding to one or two particles are coloured in grey. Oceanographic regions of the individual clusters (if existing): D: Norwegian Sea, E: Western Boundary Current region, F: Caribbean Sea, I: Irminger Current, J: Barents Sea, K: Bay of Biscay. Names of groups of clusters: Subtropical Gyre (A,F,G,E,H,K), Subpolar Gyre (B, I), Nordic Seas (D, J). b: clustering result plotted at final time. c: hierarchy of the different splits. For each split, the (inverted) y-axis shows the respective NCut after the split. Clusters corresponding to one or two particles are labelled by '.'. The initial conditions are randomly shuffled prior to plotting such that no colour dominates another one at the region boundaries.

375     To test the sensitivity of the clustering result in fig. 6 on the bin size, we applied the same method to square bins of $2°$ and $4°$, see fig. C2. The main changes to the result in fig. 6 occur in the centre and the south of the Subtropical gyre, where no specific structures were expected in the first place. Only the Bay of Biscay is not detected in the $2°$ case (fig. C2a). All other structures such as the boundary between the Subtropical and Subpolar Gyres, the Western Boundary Current region, the Caribbean Sea and the structures in the Nordic Seas are still detected very similar to the result in fig. 6.

## 5 Conclusions

We introduce a new and conceptually simple method that enables the fast construction and clustering of particle based networks to detect quasi-stationary regions with similar flow properties. Our method is based on ideas from symbolic dynamics, where a coarse but long particle itinerary can still resolve very detailed structures below the partition size. We implement a conceptually simple form of this idea and construct a bipartite graph that connects particles and bins, with links corresponding to the time-averaged conditional symbol distribution of each particle's trajectory. We use this bipartite graph to define a similarity graph on particle trajectories, to which we apply normalized cut spectral clustering. The bipartite fundament of our method enables us to use singular vectors of a related data matrix to construct a simultaneous K-way clustering solution under the normalized cut with spectral relaxation instead of computing eigenvectors of the normalized Laplacian.

Our results show that although we reduce the amount of processed data to a minimum by considering distributions over particle itineraries only, our method is powerful in handling incomplete trajectory data and is computationally efficient to implement. The basic idea of our algorithm is rooted in dynamical systems theory and symbolic dynamics, where long and coarse particle itineraries slice the state space up to scales much below the partition size. The construction of the sparse data matrix used for the singular value decomposition (SVD) has computational complexity $O(NM + N\tau)$, where $N$ is the number of particles, $M$ the number of bins and $\tau$ the number of time steps. The linear scaling with the particle number is promising for applications to large trajectory data set, although the complexity of the corresponding SVD depends on the sparsity structure of the resulting matrix, which under some parameter choices (such as very long time scales with fine binning) could become problematic.

Despite the performance and the low computational complexity of our method, the construction of the networks defined with itinerary distributions is to a certain extent ad-hoc. The construction is mostly motivated by practical requirements, i.e. the need to define a reasonable similarity measure between particles that is not too exclusive, satisfies some reasonable behaviour regarding missing data and decomposes into block-diagonal structure for invariant flow regions in ideal cases. Due to completely discarding the time dimension, our method is dependent on a fixed reference frame with respect to which the state space partition is defined. Therefore, it can not detect moving Lagrangian vortices, such as those in the 'Bickley Jet' (discussed e.g. in Hadjighasem et al. (2017)). The method presented here is the most basic way itineraries can be dealt with. When refining the definition of similarities, it is likely that graphs constructed from symbolic itineraries have a large potential for fast and reliable coherent structure detection.

For the double-gyre flow, our method successfully identifies known flow features to relatively high detail in the known transport boundaries. We demonstrate that our algorithm performs relatively well under deleting a large part of the trajectory data, making it suitable for real-world applications. We also show that an a priori low-dimensional definition of the clustering problem through a coarse binning can still detect the major flow features with an accuracy down to scales well below the bin resolution. We finally apply hierarchical clustering to the network constructed from drifter data in the North Atlantic, and successfully detect major flow regions such as the Western Boundary Current region, the Subpolar-Subtropical Gyres and the Caribbean Sea, providing the first drifter based clustering of the North Atlantic surface transport using network theory.

*Code and data availability.* All code, including the script to constrain the global drifter data to the North Atlantic, is available at github: https://github.com/OceanParcels/drifter_trajectories_network. The drifter data is publicly accessible at https://www.aoml.noaa.gov/phod/gdp/interpolated/data/all.php.

## Appendix A: Relation of $Q$ to almost-invariant sets with the transfer operator

For an introduction to finding almost-invariant sets with the transfer operator, see Dellnitz and Junge (1999), Froyland (2005) and Froyland and Padberg (2009). Assume $\tau = 2$ and denote by $C_0 = C(0)$ and $C_1 = C(1)$ the two required data matrices defined in eq. (1). The transition matrix $P$ that approximates the transfer operator from time $t = 0$ to $t = 1$ is by definition related to these matrices by: $P = D[C_0^T]^{-1} C_0^T C_1$. As described by Froyland (2005), one can find almost-invariant sets with the eigenvectors of the matrix $\hat{P} = \frac{1}{2}\left(P + \Pi^{-1} P^T \Pi\right)$, where $\Pi = \mathrm{diag}(\pi_1, \ldots, \pi_N)$ and $\pi$ corresponding to the invariant measure of the flow, i.e. $\pi P = \pi$. $\hat{P}$ can be seen as a reversible Markov chain with stationary density $\pi$, and is the random walk Laplacian $L_r[A_P]$ of the adjacency matrix

$$A_P = \frac{1}{2}\left(\Pi P + (\Pi P)^T\right). \tag{A1}$$

Froyland (2005) then proposes to find sets $\{S_1, \ldots, S_K\}$, defined by index sets $\{I_1, \ldots, I_K\}$ referring to the contained bin labels in each set, such that a generalized coherence ratio $\rho(S_1, \ldots, S_K)$ (see eq. (A2) for a definition) is maximized. We first show that this is equivalent to the NCut problem applied to $A_P$.

*Proposition 1:* Minimizing the generalized normalized cut of $A_P$ defined in eq. (6) with spectral relaxation is equal to maximizing the generalized coherence ratio $\rho$ defined by Froyland (2005).

*Proof:* By definition,

$$
\begin{aligned}
\rho(S_1, \ldots, S_K) &= \sum_{k=1}^{K} \frac{\sum_{i \in I_k, j \in I_k} \pi_i P_{ij}}{\sum_{i \in I_k} \pi_i} \\
&= \sum_{k=1}^{K} \frac{\sum_{i \in I_k, j \in I_k} A_{P,ij}}{\sum_{i \in I_k, j} A_{P,ij}} \\
&= \sum_{k=1}^{K} \frac{\sum_{i \in I_k, j} A_{P,ij} - \sum_{i \in I_k, j \notin I_k} A_{P,ij}}{\sum_{i \in I_k, j} A_{P,ij}} \\
&= K - \mathrm{NCut}(S_1, \ldots, S_K). \square
\end{aligned}
\tag{A2}
$$

Note that this relies on the fact that $\pi$ is invariant under right multiplication by $P$, from which follows that $\sum_j A_{P,ij} = \pi_i$ in the denominator of lines 2-3 in eq. (A2). We now show that the right singular vectors of $R$ defined in eq. (10) are equal to the

right eigenvectors of $\hat{P}$ if the particle measure is invariant and uniform.

440

*Proposition 2:* If the particle measure is invariant and uniform, the right eigenvectors of the matrix $\hat{P} = \frac{1}{2}\left(P + \Pi^{-1}P^T\Pi\right)$ used by Froyland (2005) are the same as the right singular vectors of $R$ defined in eq. (10) with $G = C_0 + C_1$ as defined in eq. (2).

445    *Proof:* First, note that if the particle measure is invariant and uniform in an incomplete data set, $R = \beta G$ for a constant $\beta > 0$. We therefore have to show that the eigenvectors of $G^T G$ are under the specified conditions the same as the right eigenvectors of $\hat{P}$. As the transition matrix is defined by $\Pi P = C_0^T C_1$, the symmetric adjacency matrix is $A_P := \Pi \hat{P} = \frac{1}{2}\left(G^T G - C_0^T C_0 - C_1^T C_1\right) = \frac{1}{2}G^T G - \Pi$. The last equality follows from the fact $C_0^T C_0 = C_1^T C_1 = \Pi$, as the particle measure is invariant. As $\Pi$ is assumed to be uniform, this proves the result. $\square$

450

## Appendix B: Supplementary figures double-gyre flow

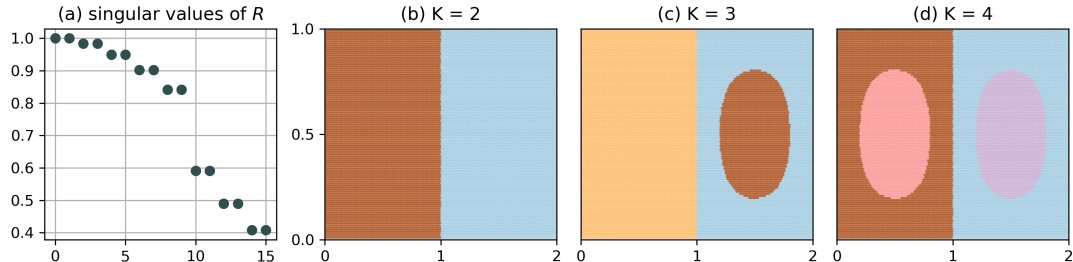

**Figure B1.** Clustering of $Q$ for the autonomous double gyre with $\Delta x = \Delta y = 0.1$, i.e. an optimal partition, and the simultaneous K-way NCut (algorithm 1 in section 3.3). The singular values are doubly degenerate as the network $Q$ consists of two disconnected sets.

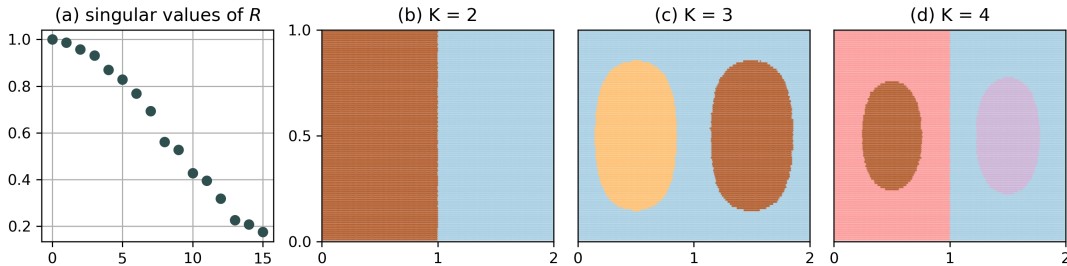

**Figure B2.** Clustering of $Q$ for the autonomous double gyre with $\Delta x = \Delta y = 0.15$, i.e. a non-optimal partition, and the simultaneous K-way NCut (algorithm 1 in section 3.3). The main separation at $x = 1$ is still resolved, but only at every even clustering step. The degeneracy of the singular values, see fig. B1, is lifted.

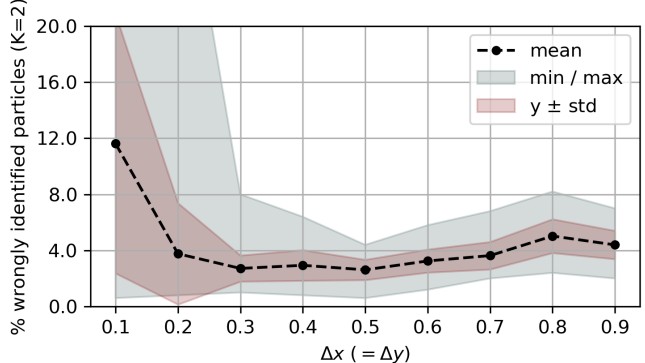

**Figure B3.** Share of incorrectly assigned particle labels for the double-gyre flow (simultaneous K-way clustering) for varying $\Delta x = \Delta y$, $K = 2$ and the incomplete (N=500) data set, compared to the baseline of the complete data set with $\Delta x = \Delta y = 0.04$, cf. fig 3. The values were obtained with 100 random incomplete data sets for each value of $\Delta x\ (= \Delta y)$. The filled areas show the minimum and maximum ranges, one standard deviation, and the points the corresponding means. It is visible that for too small bin spacing, the number of wrongly assigned labels is very high. This is most likely because the network becomes less connected, such that it is more difficult to find structure in the eigenvector corresponding to the second smallest eigenvalue of $L_s[Q]$. As explained in the main text, for bin sizes that do not exactly fit the domain, the x- and y-directions were artificially extended.

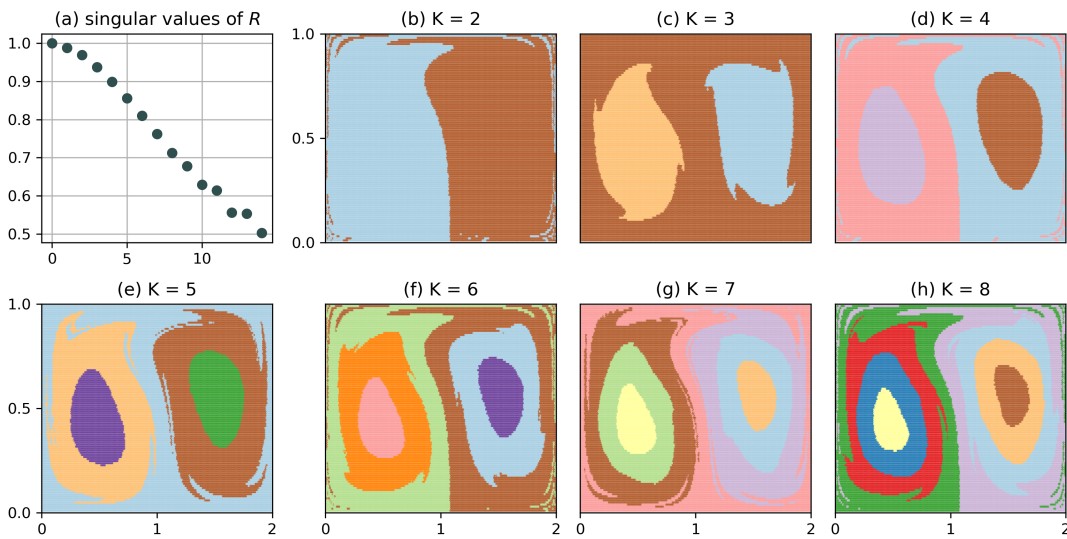

**Figure B4.** Clustering of $Q$ (simultaneous K-way clustering) for the non-autonomous double gyre with $\Delta x = \Delta y = 0.04$ and shorter trajectories of 10 gyre periods, i.e. $\tau = 101$.

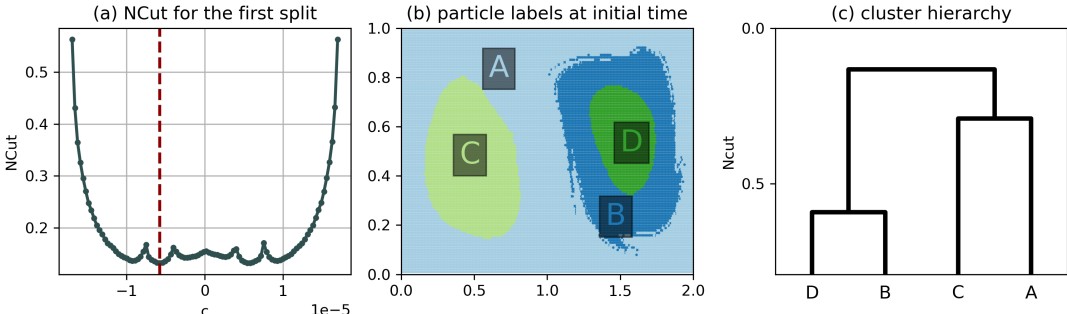

**Figure B5.** Hierarchical clustering result of the double gyre using algorithm 2 of section 3.3 and $\Delta x = \Delta y = 0.04$. a: NCut for the first split as a function of the cutoff $c$, where $c = 0$ corresponds to a cut along the transport boundary, the red line indicating the minimum found by our algorithm. b: result of the hierarchical clustering. c: dendrogram representing the hierarchy, where the horizontal lines correspond to the NCut value after each split. There is no global or local minimum at $c = 0$ in (a), which is why the transport boundary is not detected. Instead, two global minima are present in (a), but our implementation selects the left minimum for the first split due to the choice of sampling points.

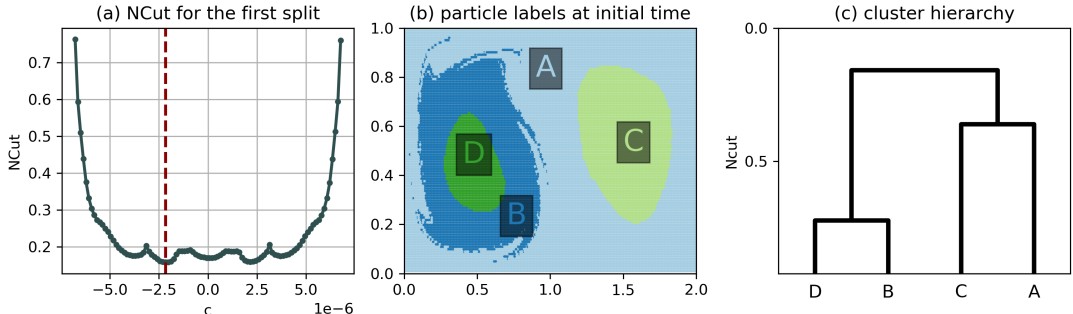

**Figure B6.** Hierarchical clustering result of the double gyre using algorithm 2 of section 3.3 and $\Delta x = \Delta y = 0.1$. a: NCut for the first split as a function of the cutoff $c$, where $c = 0$ corresponds to a cut along the transport boundary, the red line indicating the minimum found by our algorithm. b: result of the hierarchical clustering. c: dendrogram representing the hierarchy, where the horizontal lines correspond to the NCut value after each split. There is only a local minimum at $c = 0$, which is why the transport boundary is still not detected. This situation is similar to the one described by Froyland and Padberg (2009) for the transfer operator framework.

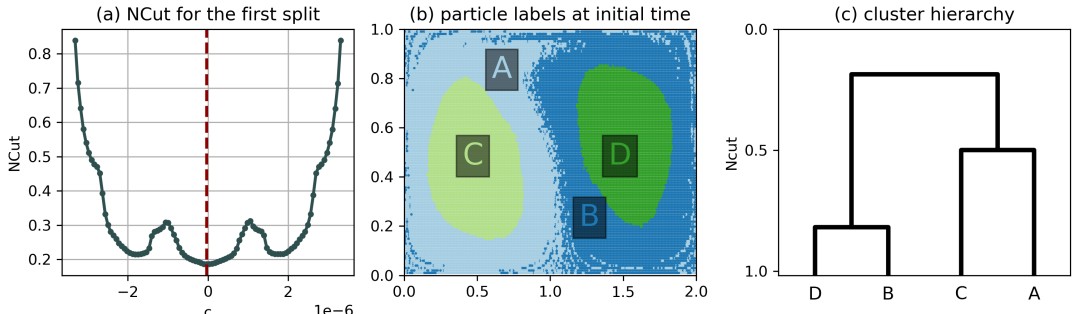

**Figure B7.** Hierarchical clustering result of the double gyre using algorithm 2 of section 3.3 and $\Delta x = \Delta y = 0.2$. a: NCut for the first split as a function of the cutoff $c$, where $c = 0$ corresponds to a cut along the transport boundary. b: result of the hierarchical clustering. c: dendrogram representing the hierarchy, where the horizontal lines correspond to the NCut value after each split. There is a global minimum at $c = 0$, so that the transport boundary is detected as the first split.

**Appendix C: Supplementary figures ocean drifters**

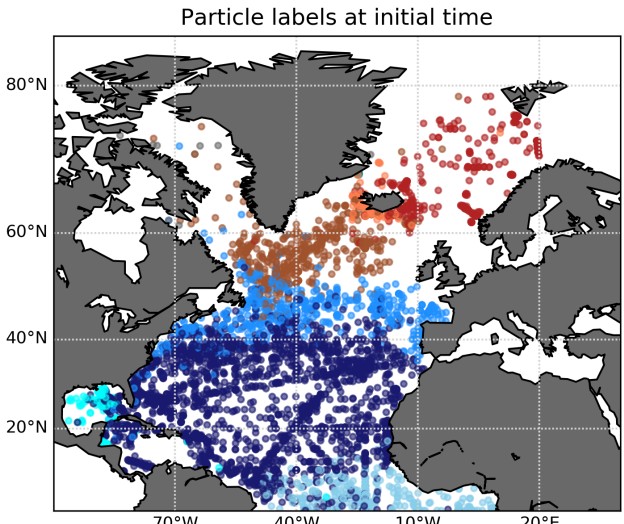

**Figure C1.** Clustering of $Q$ for the drifter data set with $\Delta x = \Delta y = 1°$ and no restriction on trajectory length, plotted at the drifter release location. Some main features as seen in fig. 6 are still visible, but some details disappear. The hierarchical clustering was stopped at a NCut value comparable to the maximum NCut in fig. 6, i.e. about 3.8.

*Author contributions.* DW performed the analysis, with support from CK, EvS and HD. DW wrote the manuscript and all authors jointly edited and revised it

*Competing interests.* The authors declare no competing interests

455

*Acknowledgements.* David Wichmann, Christian Kehl and Erik van Sebille are supported through funding from the European Research Council (ERC) under the European Union Horizon 2020 research and innovation programme (grant agreement No 715386).

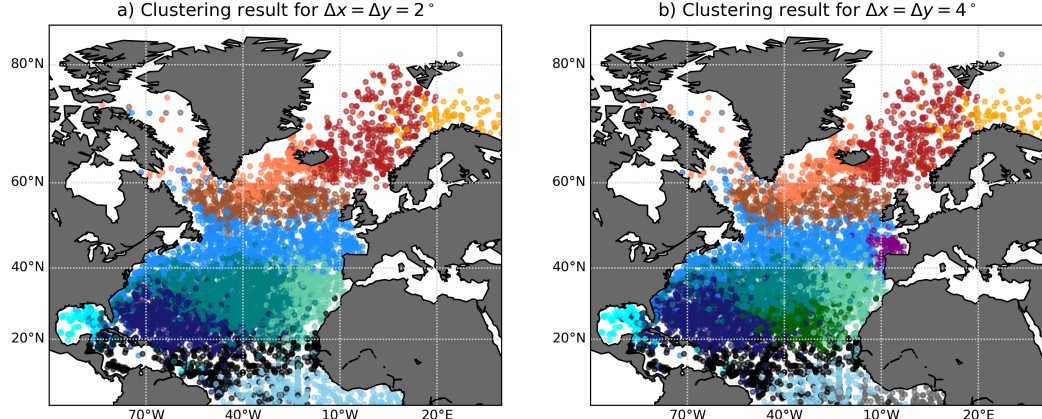

**Figure C2.** Clustering of $Q$ for the drifter data set and different bin sizes, with trajectory lengths restricted to one year, 20 clusters as in fig. 6, plotted at initial time and the drifter release location. a: $\Delta x = \Delta y = 2°$. b: $\Delta x = \Delta y = 4°$. The main features of fig. 6 are still visible, although some structures change: the Bay of Biscay is not detected in (a), and the structure of the clusters in the centre of the Subtropical Gyre change compared to fig. 6. There are also new clusters in black (a, b) and dark green (b). Note that the main features of the North Atlantic Ocean such as the Subtropical-Subpolar Gyre boundary, the Western Boundary Current region, the Caribbean Sea and the Nordic Seas are still detected, see the result in fig. 6.

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
