# Peer review of "Detecting flow features in scarce trajectory data using networks derived from symbolic itineraries: an application to surface drifters in the North Atlantic"

_Nonlinear Processes in Geophysics, 2020_

## Referee Comment (RC1) · Anonymous Referee #1 · 26 Jun 2020

The authors describe a method to cluster trajectory data in fluid flows, so that similar trajectories become identified and lead to a partition of the fluid domain, suitable for identifying coherent structures. The method is applied to the double-gyre model system an to drifter data in the North Atlantic.

The method seems to be powerful, although an exhaustive comparison with other available methodologies has not been performed. The most interesting feature is the robustness no missing data, which is clearly of interest in oceanography. The major limitation of the method seems to be the discard of any time-ordering information in the visited

locations. There are some comments about this in the manuscript, which I found sufficient.

I find the paper useful to the community (although perhaps written in a too mathematical language) and I would recommend publication provided the authors address the following minor points:

- The method share many aspects with standard spectral clustering methods (e.g. Fiedler's). One of the known limitations of these methods is that they partition the network in 'balanced' (i.e. not too different sizes) parts. This is usually an advantage in image processing and in computer-load redistribution, but I find this an important limitation in the present application to fluid flows. I ask the authors to state if this is a limitation of the present method and its possible impact on applications.

- Is there any criterion to determine an 'optimum' number K of network parts, or when to stop the iterative hierarchical partitioning?

- In the application to the North-Atlantic drifter data set, the authors declare to look for 'rigid, stationary features'. Nevertheless, the particle position at the beginning of the trajectories and at the end (Figs 6a and 6b) are different. Could you discuss the implications of this on the 'stationarity' of the structures and in relationship with trajectory duration?

- Could you comment on the reasons for the change in size of the ellipsoidal structures identified in Fig. 4 with respect to the ones in Fig. 3?

- The set S in the denominator of Eq. (6) is not defined.

- the authors use the word 'similar' in lines 172, 341 and 350 in an unclear meaning, specially because in other parts of the manuscript some 'similarity' measures are defined and used. Please use 'equivalent' o 'equal' if this is the intended meaning of 'similar' there, of choose a more precise word if it is not.

---

## Referee Comment (RC2) · Anonymous Referee #2 · 5 Jul 2020

The authors introduce a novel method for the identification of almost-invariant sets in fluid flows based on sparse and incomplete trajectory. By binning the particle positions and ignoring the temporal information a bipartite network is obtained that connects particles and bins. Links are weighted with the probability that the particle can be found in that bin. By solving either a standard or hierarchical NCut clustering problem, dominant flow features are detected from groups of trajectories that behave in a similar manner. The approach is successfully applied to the double-gyre benchmark flow and to surface drifter data in the North Atlantic and shown to perform well for the respective

setting, even in the case of scarce trajectory data. The paper is very well and clearly written and the method is certainly of great interest to the readers of NPG. However, a moderate revision addressing the points outlined below is required before the paper can be accepted for publication. My major criticism is that a more exhaustive study of the method is required.

Major points:

- A new and potentially useful method is introduced and discussed in relation to established methods, but there is no direct comparison, neither w.r.t. to the resulting clustering nor to computational run-times. While a detailed comparative study is certainly beyond the scope of this paper, the discussion should be extended in that respect. What are the advantages and what are the limitations of the method – in comparison to the established approaches?

- The two case studies (double-gyre and drifters) are each treated with a different clustering approach (K-way clustering vs hierarchical clustering) and it remains open how these two choices influence the results, in particular as there is no obvious spectral gap in the double-gyre system indicating an appropriate choice of K. I also assume that the results depend very much on the trajectory length but this is only briefly mentioned for the drifter data (l. 286). These points could be addressed in a more detailed study of the double-gyre flow, taking into account different flow times and the two clustering approaches.

- I don't understand how figure 7 relates to the hierarchical clustering that is carried out for the drifter data and where the indicated separations between the different geographical regions come from. Does fig 7 show the results for the K-way clustering? Some more explanations are required in my view.

- The chosen bin size of 0.4 for the sparse data case (l. 246) means that some bins have to cropped in order to fit into the domain and as a result the bins are not equally sized. How is that done and how does that influence the computation?

Minor points:

- Probably not all readers are familiar with the concept of almost-invariant sets, so this should be briefly motivated in the introduction.

- The other anonymous referee already pointed out that S is not defined in the denominator of equation (6). In that context it would be helpful for the reader if the authors briefly explain the cost function.

- I also realized that the concept of "similarity" is used in a rather sloppy sense.

- The color figures are not appropriate for gray-scale printouts.

- I suggest some critical proof-reading (e.g. capitalization in reference list).

---

## Author Comment (AC2)

**Answers to the comments of Referee 2**

**Note: Line numbers refer to the track-changed version**

**Comment 1**

A new and potentially useful method is introduced and discussed in relation to established methods, but there is no direct comparison, neither w.r.t. to the resulting clustering nor to computational run-times. While a detailed comparative study is certainly beyond the scope of this paper, the discussion should be extended in that respect. What are the advantages and what are the limitations of the method – in comparison to the established approaches?

**Answer to comment 1**

Thank you very much for that comment. A more detailed comparison was indeed lacking in the first version. In the revised version, we will add a subsection (3.4) in the methods section to point out the differences to previous methods, advantages and limitations.

**Changes in text, line 236:**

**3.4 Comparison to existing methods**

Our method aims to detect groups of particles, with trajectories of different groups having only little overlap. In this sense, our method detects groups of particles with little mixing between each other, which is close to detecting almost-invariant sets according to Froyland (2005). Yet our method is different from detecting almost-invariant sets with the transfer operator in

- 240 several aspects. First, it is based on similarities between individual particles rather than spatial sets (bins), which allows us to cluster on the particle level rather than the bin level. As we will show in section 4.1, this can be used to resolve flow features down to scales much below the bin size. Secondly, our method employs the full trajectory information in terms of a particle's symbolic itinerary, rather than just the start and end points or symbols. In practical applications, this can be an advantage compared to the transfer operator framework, as there is no need in assuming Markovian behaviour of the flow given a state
- 245 space partition, as done by e.g. Froyland et al. (2014). There are also major differences between our method and other existing methods that cluster on the particle level (Froyland and Padberg-, First, these methods only compare particles at equal times, while we disregard the time information. This can be a significant advantage in situations where the major features of a flow are approximately stationary, i.e. can be seen as part of a (noisy) autonomous dynamical system. In this case, using the time information of drifter trajectories should not be necessary. Especially
- 250 for the ocean drifter dataset, containing drifters of different starting times and lengths, it would be very difficult if not impossible to find sub-basin large scale structures when restricting to drifters that necessarily overlap temporally, although this is possible on the global scale to identify the basins themselves (Froyland and Padberg-Gehle, 2015; Banisch and Koltai, 2017). Note that simply placing all drifters at the same initial time and proceeding with one of the existing methods would lead to further problems, as there is ambiguity in which point of a trajectory should be taken for the initial time, probably requiring more data

255 processing such as demanding an initially uniform particle distribution. Our method of simplifying the trajectories does not have these problems by construction, and can be readily applied to scarce drifter datasets. From a computational perspective, setting up one of the sparse matrices C(t) in eq. (1) is O(N) such that computing the matrix G in eq. (2) is  $O(N\tau)$ . Computing  $d[G^T]$  is O(NM) as is the product  $Gd[G^T]$ ). In total, computing R of eq. (10) is therefore of computational complexity  $O(NM + N\tau)$ . If we work with R directly, i.e. we use the simultaneous K-way clustering method

260 described in algorithm 1 in section 3.3, computing this network is of lower computational complexity as the computation of the networks used by other studies (Padberg-Gehle and Schneide, 2017; Banisch and Koltai, 2017; Hadjighasem et al., 2016) . These methods typically rely on comparing particle positions between all particles at all time instances, i.e. they scale with  $O(N^2\tau)$  in the worst case, although a nearest-neighbour search as applicable to the studies of Padberg-Gehle and Schneide (2017) and Banisch and Koltai (2017) can reduce the  $N^2$  term to something like  $N \log N$ . Further, the matrix R in eq. (10) is sparser

- than  $L_s[Q]$  or can have column dimension (= number of bins M) significantly lower than row dimension (= number of particles N), cf. section 4.1. In these cases, computing the SVD of R instead of the eigenvectors of  $L_s$  can lead to computational speed up. Finally, it is interesting to note that the computation of the network is faster for coarser partitions, i.e. when particles are connected in the network even when their trajectories are far apart, as the number of bins M decreases. This is opposite to the methods of Padberg-Gehle and Schneide (2017) and Banisch and Koltai (2017), where computing the network becomes more
- 270 costly for larger spatial scale parameters (called c in both studies). The major drawback of our method is the dependence on a reference frame with respect to which the phase space partition and thus the symbolic itineraries are defined. This can be understood when imagining a time-independent flow from a rotating reference frame. The rotation of the reference frame contributes to a particle's itinerary, and, by averaging over different points in time, non-zero similarities between trajectories can result from the sole rotation of the reference frame. Due to this reason,
- 275 our method can not be applied to strongly time-dependent systems such as the Bickley jet model flow where coherent vortices are transported in a periodic background flow. It is, however, still possible to detect transport boundaries in time-dependent flows such as the periodically driven double-gyre flow, as we show in section 4.1, where particle trajectories belonging to different invariant sets can still be distinguished with a fixed partition.

**Comment 2**

The two case studies (double-gyre and drifters) are each treated with a different clustering approach (K-way clustering vs hierarchical clustering) and it remains open how these two choices influence the results, in particular as there is no obvious spectral gap in the double-gyre system indicating an appropriate choice of K. I also assume that the results depend very much on the trajectory length but this is only briefly mentioned for the drifter data (I. 286). These points could be addressed in a more detailed study of the double-gyre flow, taking into account different flow times and the two clustering approaches.

**Answer to comment 2**

Thank you very much for this suggestion. It, in fact, helped us to better understand our method. We now applied also the hierarchical clustering method to the double-gyre flow and found that there is not necessarily a global minimum in the NCut for the first split (along the transport boundary). We found that such a minimum appears only for larger bin sizes, see the figures below (B5-B7). The lack of a global minimum for this system was found before for the transfer operators by Froyland & Padberg (2009), and is partially also a consequence of the very idealized flow, where the objective function is symmetric around c=0. Nevertheless, because of this sensitivity on the bin size, we now also computed the clustering results for the North Atlantic drifters for different bin sizes, see the figure below (C2). We do not find any striking difference in the main features of the North Atlantic Ocean. The text will be changed accordingly.

Changes in text, line 328:

We also tested the algorithm for shorter trajectories, cf. fig. B4 in the appendix, showing an expected change of the boundary filaments between the left and right sides of the fluid, which mix less in the shorter period of time. To better understand

- the differences between algorithms 1 and 2 introduced in section 3.3, we also applied the hierarchical NCut method to the non-autonomous double gyre flow. A problem arises here for the first split into two clusters, as there is no unique minimum in the objective function in eq. (6), i.e determining the cutoff *c* in algorithm 2, H5 (cf. section 3.3), is ambiguous for Δx = Δy = 0.04. The lack of a unique minimum for the NCut has been observed before for the same model flow by Froyland and Padberg (2009) in the transfer operator framework (see their fig. 15) corresponding to the lack of a unique maximum of the coherence ratio there (see proposition 1 in appendix A). Yet, for Δx = Δy = 0.04, in our case, the split between the left and right sides along the transport boundary (the K = 2 split in fig. 3) does not even have a local minimum
- for the NCut, cf. fig. B5, as opposed to the local maximum of Froyland and Padberg (2009). This however changes to a local minimum for Δx = Δy = 0.1 (fig. B6) and finally to a global minimum for Δx = Δy = 0.2 (fig. B7). A possible explanation of this dependence on bin size is that the addition of noise (i.e. larger bins) decreases the coherence of the gyre centres compared
  to the transport boundary, making the latter easier to be detected. Due to the sensitivity of the hierarchical clustering result to
  - the bin size, we test different bin sizes for the clustering of the North Atlantic drifters in section 4.2 (cf, fig. C2).

**Figure B5.** Hierarchical clustering result of the double gyre using algorithm 2 of section 3.3 and  $\Delta x = \Delta y = 0.04$ , a: NCut for the first split as a function of the cutoff *c*, where *c* = 0 corresponds to a cut along the transport boundary, the red line indicating the minimum found by our algorithm, b: result of the hierarchical clustering, c: dendrogram representing the hierarchy, where the horizontal lines correspond to the NCut value after each split. There is no global or local minimum at *c* = 0 in (a), which is why the transport boundary is not detected. Instead, two global minima are present in (a), but our implementation selects the left minimum for the first split due to the choice of sampling points.

**Figure B6.** Hierarchical clustering result of the double gyre using algorithm 2 of section 3.3 and  $\Delta x = \Delta y = 0.1$ , a: NCut for the first split as a function of the cutoff *c*, where c = 0 corresponds to a cut along the transport boundary, the red line indicating the minimum found by our algorithm, b: result of the hierarchical clustering, c: dendrogram representing the hierarchy, where the horizontal lines correspond to the NCut value after each split. There is only a local minimum at c = 0, which is why the transport boundary is still not detected. This situation is similar to the one described by Froyland and Padberg (2009) for the transfer operator framework.

Figure B7. Hierarchical clustering result of the double gyre using algorithm 2 of section 3.3 and  $\Delta x = \Delta y = 0.2$ , a: NCut for the first split as a function of the cutoff c, where c = 0 corresponds to a cut along the transport boundary. b: result of the hierarchical clustering, c: dendrogram representing the hierarchy, where the horizontal lines correspond to the NCut value after each split. There is a global minimum at c = 0, so that the transport boundary is detected as the first split.

---

## Author Response (AR1)

**Answers to the comments of Referee 1**

**Note: Line numbers refer to the track-changed version**

**Comment 1**

"The method shares many aspects with standard spectral clustering methods (e.g. Fiedler's). One of the known limitations of these methods is that they partition the network in 'balanced' (i.e. not too different sizes) parts. This is usually an advantage in image processing and in computer-load redistribution, but I find this an important limitation in the present application to fluid flows. I ask the authors to state if this is a limitation of the present method and its possible impact on applications."

**Answer to comment 1**

Thank you for this comment. Indeed, the NCut finds more balanced clusters in terms of their weighted size. This is also why this kind of clustering is expected to fail when looking for very small structures compared to the fluid domain, e.g. for ocean eddies. For our application however, this is not a limitation because we are looking for large scale structures in the North Atlantic Ocean. We will include this in the methods section.

**Changes in text, line 164:**

*Minimizing the NCut leads to a clustering result that tries to balance the different terms in eq. (6) such that the resulting clusters are of approximately equal size in terms of their total relative weight (Shi & Malik 2000). While this poses no serious problem for detecting large scale flow features in the ocean, it is certainly a limitation for detecting even smaller structures such as eddies or jets in a large ocean domain, and we explicitly exclude such examples from the scope of our method.*

**Comment 2**

"Is there any criterion to determine an 'optimum' number K of network parts, or when to stop the iterative hierarchical partitioning?"

**Answer to comment 2**

There is no general criterion how to truncate the spectral embedding. A heuristic was proposed in different contexts (also fluid dynamics) to look for the largest spectral gap. But this might not work for many applications. We will include a more detailed discussion about this issue in the methods section.

**Changes in text, line 228:**

*Note that there is no general rule to determine the number of clusters K in algorithm 1, or where to stop the hierarchical clustering procedure in algorithm 2. A popular heuristic to determine K is to look for a prominent gap in a spectrum of $L\_s$ and choose K as the number of smallest eigenvalues before that gap (Hadjighasem et al. 2016). This is however problematic for systems with no prominent spectral gap, which is the case for the systems considered here (cf. section 4). For algorithm 1, we therefore compute the clustering results for different values of K to see how the results depend on this choice. For algorithm 2, one can set a maximum value on the cost function in*

*eq. (6) as suggested by Shi & Malik (2000). Yet, the cost function is rather abstract, and there is no general rule what this value should be. Here, we compute the clusters up to a certain (arbitrary) order and compare the results to known structures in oceanography.*

**Comment 3**
In the application to the North-Atlantic drifter data set, the authors declare to look for rigid, stationary features. Nevertheless, the particle position at the beginning of the trajectories and at the end (Figs 6a and 6b) are different. Could you discuss the implications of this on the 'stationarity' of the structures and in relationship with trajectory duration?

**Answer to comment 3**
Thank you for this comment. Indeed we were not clear on the fact that the trajectories of the different particle groups do have non-zero overlap (no perfect clustering). As a result, the initial positions of particles of one cluster can overlap with the final positions of another cluster, i.e. the spatial extent of the cluster is only approximately stationary. We will include some more clarification on this issue in the results section.

**Changes in text, line 362:**
*Note here that the trajectories in the different clusters do have small but non-zero overlap, such that the spatial extent of the clusters can be different at initial and final time.*

**Comment 4**
Could you comment on the reasons for the change in size of the ellipsoidal structures identified in Fig. 4 with respect to the ones in Fig. 3?

**Answer to comment 4**
Thanks for this comment. This question is quite difficult to answer, especially as there are many steps involved in the network construction and the clustering that are not obvious or where it is difficult to get an intuition for. Increasing the bin size will lead to more trajectory overlaps, but what that means directly for the clusters resulting from k-Means on the eigenvectors is difficult to understand. For us, having a decrease in the size of these structures indicates that increasing the bin size does not preserve all structures, but only the most dominant ones (i.e. the separation between left and right of the fluid domain). But we were not specific enough about this in the previous version. We will add more explanation about this point in the new version.

**Changes in text, line 305:**
*The corresponding clustering result still resolves the most dominant structure up to very high resolution: the split between the left and right side is preserved, and even the structure of the transport barrier is very similar to the one in fig. 3. The results for K=3 and K=4 are however different from fig. 3, as the gyre centres appear smaller. This indicates that only the most prominent structures, here the separation between the left and right sides, are preserved under coarsening the partition. Nevertheless, figs. 4c-d do still give an impression about the flow structures at higher orders,*

*though not completely equal to the high resolution case. Note that the singular values (fig. 4a) are strongly suppressed compared to the M=1,250 case in fig. 3.*

**Comment 5**
The set S in the denominator of Eq. (6) is not defined.

**Answer to comment 5**
Thank you for noting. We will do so.

**Changes in text, line 155:**
*Assume we are given an undirected network defined on a discrete set S containing N vertices, with edges given by the symmetric adjacency matrix $Q \in \mathbb{R}^{N \times N}$.*

**Comment 6**
"The authors use the word 'similar' in lines 172, 341 and 350 in an unclear meaning, especially because in other parts of the manuscript some 'similarity' measures are de-fined and used. Please use 'equivalent' or 'equal' if this is the intended meaning of 'similar' there, of choose a more precise word if it is not."

**Answer to comment 6**
Thank you for noting. Indeed we were inconsistent in the usage of 'similar' and imprecise. We will change this in the revised version.

**Changes in text, line 185:**
*... it follows that these eigenvectors are equal to the left singular vectors corresponding to...*

**Changes in text, line 438:**
*Minimizing the generalized normalized cut of $A_P$ defined in eq. (6) with spectral relaxation is equal to maximizing the generalized coherence ratio*

**Changes in text, line 447:**
*We now show that the right singular vectors of R defined in eq. (10) are equal to the right eigenvectors of $\hat{P}$ if the particle measure is invariant and uniform.*

**Answers to the comments of Referee 2**

**Note: Line numbers refer to the track-changed version**

**Comment 1**

A new and potentially useful method is introduced and discussed in relation to established methods, but there is no direct comparison, neither w.r.t. to the resulting clustering nor to computational run-times. While a detailed comparative study is certainly beyond the scope of this paper, the discussion should be extended in that respect. What are the advantages and what are the limitations of the method – in comparison to the established approaches?

**Answer to comment 1**

Thank you very much for that comment. A more detailed comparison was indeed lacking in the first version. In the revised version, we will add a subsection (3.4) in the methods section to point out the differences to previous methods, advantages and limitations.

**Changes in text, line 236:**

**3.4 Comparison to existing methods**

Our method aims to detect groups of particles, with trajectories of different groups having only little overlap. In this sense, our method detects groups of particles with little mixing between each other, which is close to detecting almost-invariant sets according to Froyland (2005). Yet our method is different from detecting almost-invariant sets with the transfer operator in several aspects. First, it is based on similarities between individual particles rather than spatial sets (bins), which allows us to cluster on the particle level rather than the bin level. As we will show in section 4.1, this can be used to resolve flow features down to scales much below the bin size. Secondly, our method employs the full trajectory information in terms of a particle's symbolic itinerary, rather than just the start and end points or symbols. In practical applications, this can be an advantage compared to the transfer operator framework, as there is no need in assuming Markovian behaviour of the flow given a state space partition, as done by e.g. Froyland et al. (2014).

There are also major differences between our method and other existing methods that cluster on the particle level (Froyland and Padberg- . First, these methods only compare particles at equal times, while we disregard the time information. This can be a significant advantage in situations where the major features of a flow are approximately stationary, i.e. can be seen as part of a (noisy) autonomous dynamical system. In this case, using the time information of drifter trajectories should not be necessary. Especially for the ocean drifter dataset, containing drifters of different starting times and lengths, it would be very difficult if not impossible to find sub-basin large scale structures when restricting to drifters that necessarily overlap temporally, although this is possible on the global scale to identify the basins themselves (Froyland and Padberg-Gehle, 2015; Banisch and Koltai, 2017). Note that simply placing all drifters at the same initial time and proceeding with one of the existing methods would lead to further problems, as there is ambiguity in which point of a trajectory should be taken for the initial time, probably requiring more data processing such as demanding an initially uniform particle distribution. Our method of simplifying the trajectories does not have these problems by construction, and can be readily applied to scarce drifter datasets.

From a computational perspective, setting up one of the sparse matrices $C(t)$ in eq. (1) is $O(N)$ such that computing the matrix $G$ in eq. (2) is $O(N\tau)$. Computing $d[G^T]$ is $O(NM)$ as is the product $Gd[G^T]$). In total, computing $R$ of eq. (10) is therefore of computational complexity $O(NM + N\tau)$. If we work with $R$ directly, i.e. we use the simultaneous K-way clustering method described in algorithm 1 in section 3.3, computing this network is of lower computational complexity as the computation of the networks used by other studies (Padberg-Gehle and Schneide, 2017; Banisch and Koltai, 2017; Hadjighasem et al., 2016) . These methods typically rely on comparing particle positions between all particles at all time instances, i.e. they scale with $O(N^2\tau)$ in the worst case, although a nearest-neighbour search as applicable to the studies of Padberg-Gehle and Schneide (2017) and Banisch and Koltai (2017) can reduce the $N^2$ term to something like $N \log N$. Further, the matrix $R$ in eq. (10) is sparser
* * *
**Comment 2**

The two case studies (double-gyre and drifters) are each treated with a different clustering approach (K-way clustering vs hierarchical clustering) and it remains open how these two choices influence the results, in particular as there is no obvious spectral gap in the double-gyre system indicating an appropriate choice of K. I also assume that the results depend very much on the trajectory length but this is only briefly mentioned for the drifter data (l. 286). These points could be addressed in a more detailed study of the double-gyre flow, taking into account different flow times and the two clustering approaches.

**Answer to comment 2**

Thank you very much for this suggestion. It, in fact, helped us to better understand our method. We now applied also the hierarchical clustering method to the double-gyre flow and found that there is not necessarily a global minimum in the NCut for the first split (along the transport boundary). We found that such a minimum appears only for larger bin sizes, see the figures below (B5-B7). The lack of a global minimum for this system was found before for the transfer operators by Froyland & Padberg (2009), and is partially also a consequence of the very idealized flow, where the objective function is symmetric around c=0. Nevertheless, because of this sensitivity on the bin size, we now also computed the clustering results for the North Atlantic drifters for different bin sizes, see the figure below (C2). We do not find any striking difference in the main features of the North Atlantic Ocean. The text will be changed accordingly.

**Changes in text, line 328:**

[revised manuscript text omitted]

**Comment 3**

I don't understand how figure 7 relates to the hierarchical clustering that is carried out for the drifter data and where the indicated separations between the different geographical regions come from. Does fig 7 show the results for the K-way clustering? Some more explanations are required in my view.

**Answer to Comment 3**

Thanks a lot for this comment. You were absolutely right that the figure had little meaning in the context of hierarchical clustering. We will remove it in the revised manuscript.

**Comment 4**

The chosen bin size of 0.4 for the sparse data case (l. 246) means that some bins have to cropped in order to fit into the domain and as a result the bins are not equally sized. How is that done and how does that influence the computation?

**Answer to comment 4**

Thanks for that comment. Indeed the bins are not equally sized then, or we artificially extend the domain to y = 1.2. The text will be adapted accordingly.

**Changes in text, line 316:**

the bin size such that the network becomes connected again. Therefore, we choose $\Delta x = \Delta y = 0.4$ . In doing so, we effectively extend the domain in the $y$-direction to $y = 1.2$ and disregard the fact that the top row of bins is not completely covered with initial conditions. Figure 5 shows the result for the clustering of $Q$ for the incomplete data set, plotted on top of

**Comment 5**

Probably not all readers are familiar with the concept of almost-invariant sets, so this should be briefly motivated in the introduction.

**Answer to comment 5**

We will include a brief explanation in the introduction.

**Changes in text, line 57:**

method is naturally extendable to incomplete trajectory data and thus readily applicable to ocean drifters. Conceptually, our method is close to detecting minimally mixing fluid regions, so called almost-invariant sets (Froyland, 2005), although there are also important differences to this method, cf. section 3.4. In our case, almost-invariant sets are represented by the initial

60  conditions of groups of particles, with trajectories of different groups having only little overlap.

**Comment 6**

The other anonymous referee already pointed out that S is not defined in the denominator of equation (6). In that context it would be helpful for the reader if the authors briefly explain the cost function.

**Answer to comment 6**

Thank you, this will be fixed in the revised version.

**Changes in text, line 155**

155  Assume we are given an undirected network  defined on a discrete set $S$ containing $N$ vertices , with edges given by the symmetric adjacency matrix  $Q \in \mathbb{R}^{N \times N}$. We assume that $Q$ is connected. If it is not connected, we focus on  each connected component separately . According to Shi and Malik (2000), the normalized cut of a partition of the nodes into $K$ sets $S_1, \ldots, S_K$, $S = \cup_{i=1}^{K} S_i$, is defined as

$$\text{NCut}(S_1, \ldots, S_K) := \sum_{i}^{K} \frac{Q(S_i, S_i^C)}{Q(S_i, S)}. \tag{6}$$

160  Here, $Q(S_i, S_j)$ is the sum of all weights connecting $S_i$ and $S_j$, i.e. $Q(S_i, S)$ is the sum of all weights connected to $S_i$. $S_i^C$ denotes the complement of $S_i$. The term $\frac{Q(S_i, S_i^C)}{Q(S_i, S)}$ appearing in the definition of the NCut in eq. (6) is simply the total weight of all the edges connecting a set $S_i$ to its complement relative to the total weight of the set $S_i$. Clustering a graph according to the NCut refers to finding a partition $\{S_k\}$ such the objective function in eq. (6) is minimized. Note that for an increasing

**Comment 7**

I also realized that the concept of "similarity" is used in a rather sloppy sense.

**Answer to comment 7**
Thank you for noting, indeed we were not very inconsistent in the usage of 'similar' and imprecise. We will change this in the revised version.

**Changes in text, line 185:**
*... it follows that these eigenvectors are equal to the left singular vectors corresponding to...*

**Changes in text, line 438:**
*Minimizing the generalized normalized cut of $A_P$ defined in eq. (6) with spectral relaxation is equal to maximizing the generalized coherence ratio*

**Changes in text, line 447:**
*We now show that the right singular vectors of R defined in eq. (10) are equal to the right eigenvectors of $\hat{P}$ if the particle measure is invariant and uniform.*

**Comment 8**
The color figures are not appropriate for gray-scale printouts

**Answer to comment 8**
Thanks for the comment. As we need to use quite a few colours to illustrate the clusters, and as papers in NPG are available online for free, we do not think this is a major problem and would like to leave this issue to the Editor.

**Comment 9**
I suggest some critical proof-reading (e.g. capitalization in reference list).

**Answer to comment 9**
Thank you for also checking the references so carefully. We have gone through them again and will make appropriate changes, also regarding capitalization and doi's in the references.

**References**

[revised manuscript text omitted]

---

## Author Response (AR2)

**Response to Editor's comments**

Note: Line numbers refer to the track changed version.

**Remark:** In addition to the suggested changes, we have made a minor change in section 4.2. We previously referred to cluster "I" as Greenland Current. After internal feedback, we decided to call it Irminger Current, which seems to be more accurate regarding the cluster location and spatial extent.

**Comment 1**
p5, eq (2): \tau-1 instead of \tau in summation; same in line 109

**Answer to comment 1**
Thank you for noting! We have corrected it in the new version

**Changes in text, eq. (2) and line 109:**

$$G = \sum_{t=0}^{\tau\ \tau-1} C(t). \tag{2}$$

The matrix $G$ has a simple interpretation: $G_{nm}$ is equal to the number of times that particle $n$ visited bin $m$ at the time instances $0,1,\ldots,\tau$ $0,1,\ldots,\tau-1$. $G$ defines a bipartite graph, i.e. a graph connecting objects of different type, here particles and bins. In the following, we define the degree vector $d[A] \in \mathbb{R}^h$ of an arbitrary matrix $A \in \mathbb{R}^{h \times l}$ by $d[A]_i := \sum_{j=1}^l A_{ij}$, and the degree matrix $D[A] \in \mathbb{R}^{h \times h}$ as $D[A] := \mathrm{diag}(d[A])$.

**Comment 2**
p9, l207: subscript n instead of N in S_2

**Answer to comment 2**
Thanks for checking so thoroughly! We have corrected this in the new version.

**Changes in text, line 207- 208:**

H5 Find a cutoff $c$ such that the sets of nodes defined by $S_1 = \{n \in \{1,\ldots,N\} \mid v_{r,1,n} < c\}$ and $S_2 = \{n \in \{1,\ldots,N\} \mid v_{r,1,N} \geq c\}$ $S_2 = \{n \in \{1,\ldots,N\} \mid v_{r,1,n} \geq c\}$ minimize the NCut for the two sets $S_1$ and $S_2$.

H6 Split the original network into two networks, with respective adjacency matrices $Q_{S_1}, Q_{S_2}$ defined by the projection of $Q$ onto these sets.

**Comment 3**
p11, l283 "i.e. i.e"

**Answer to comment 3**
Thanks! We have corrected this in the new version.

**Changes in text, line 283:**

where $f(x,t) = \epsilon \sin(\omega t)x^2 + (1 - 2\epsilon \sin(\omega t))x$, and $A = 0.25$, $\epsilon = 0.25$ and $\omega = 2\pi$. Similar to Banisch and Koltai (2017), we initially place 20,000 particles on the vertices of a uniform grid on the domain $(0,2) \times (0,1)$ and compute trajectory outputs for twenty gyre periods with time steps of 0.1, i.e.  we have $\tau = 201$. The eigenvectors are computed with the SVD of the matrix $R$ in eq. (10). These eigenvectors are then used for the K-way clustering algorithm with k-Means, cf. algorithm 1 in
285   section 3.3.

**Comment 4**

In sect. 4.2, the authors discard some trajectories that do not belong to the largest connect component. How many trajectories are discarded and how does that depend on the bin size? Can anything be said about the geographical location of these trajectories?

**Answer to comment 4**

Thank you for this question. Previously we indeed did not look at the number of trajectories / their locations. We have now done so, and only seven trajectories are deleted, six of which are sending erroneous data. The other trajectory is short and at the straight of Gibraltar.

**Changes in text, line 342**

[revised manuscript text omitted]